# Advances of Zinc Signaling Studies in Prostate Cancer

**DOI:** 10.3390/ijms21020667

**Published:** 2020-01-19

**Authors:** Dangdang Li, Daniel B. Stovall, Wenmeng Wang, Guangchao Sui

**Affiliations:** 1Key Laboratory of Saline-alkali Vegetation Ecology Restoration, Ministry of Education, College of Life Science, Northeast Forestry University, Harbin 150040, China; lidd@nefu.edu.cn (D.L.); wangwenmeng@nefu.edu.cn (W.W.); 2College of Arts and Sciences, Winthrop University, Rock Hill, SC 29733, USA; stovalld@winthrop.edu

**Keywords:** zinc, prostate cancer, signaling pathway, proliferation, apoptosis, metastasis, zinc transporter, zinc finger, cancer therapy

## Abstract

Prostate cancer (PCa) is one of the most common cancers and the second leading cause of cancer-related death among men worldwide. Despite progresses in early diagnosis and therapeutic strategies, prognosis for patients with advanced PCa remains poor. Noteworthily, a unique feature of healthy prostate is its highest level of zinc content among all soft tissues in the human body, which dramatically decreases during prostate tumorigenesis. To date, several reviews have suggested antitumor activities of zinc and its potential as a therapeutic strategy of PCa. However, an overview about the role of zinc and its signaling in PCa is needed. Here, we review literature related to the content, biological function, compounds and clinical application of zinc in PCa. We first summarize zinc content in prostate tissue and sera of PCa patients with their clinical relevance. We then elaborate biological functions of zinc signaling in PCa on three main aspects, including cell proliferation, death and tumor metastasis. Finally, we discuss clinical applications of zinc-containing compounds and proteins involved in PCa signaling pathways. Based on currently available studies, we conclude that zinc plays a tumor suppressive role and can serve as a biomarker in PCa diagnosis and therapies.

## 1. Introduction

Prostate cancer (PCa) is the most commonly diagnosed cancer and has the second highest death rate among men worldwide, particularly in developed countries [1,2]. For example, in the United States, the estimated new PCa cases are 174,650, and estimated PCa-related deaths are 31,620 in 2019 [3]. With the increased life expectancy of humans, the incidence of PCa is expected to accelerate [4]. Thus, early diagnosis to prevent the onset of this disease and treatment to delay tumor progression are urgently needed in clinic. In early detection, the likelihood and progressive phenotype of PCa are heavily reliant on prostate-specific antigen (PSA)-based screening [1,4,5,6]. However, this method can lead to overdiagnosis, unnecessary biopsies and overtreatment of patients who may have only needed active surveillance [7,8]. In clinical applications, the most common and widely accepted therapeutic options for PCa patients are the combinations of surgery, radiotherapy and hormone therapy [9,10,11]. Unfortunately, these treatments could lead to side effects and considerable recurrence rates [9]. Therefore, it is necessary to develop novel strategies to prevent, diagnose and effectively treat PCa patients in clinic.

Remarkably, accumulating evidence has proven that various factors, including diet, lifestyle and genetic background, are closely related to high morbidity rates of PCa [12,13,14]. Metal ions are important components in food and play vital roles in maintaining human health [15,16]. Zinc, in particular, has been a nutrient of great interest, given that it acts as not only a cofactor for the functions of over 300 enzymes but also an essential component for zinc finger (ZF)-containing transcription factors (TFs), copper/zinc superoxide dismutase and various proteins involved in DNA repair [17,18,19,20]. Generally, zinc homeostasis is tightly regulated by ion channels and zinc transporters, including the Zrt-Irt-like protein (ZIP) and zinc transporter (ZnT) families [21,22,23]. More recently, zinc was recognized as an intra- and intercellular signaling mediator with vital regulatory functions in signal transduction [18,24]. Based on those studies, zinc plays a crucial role in various biological events, including antioxidant defense, DNA repair, tissue repair, immunoprotection, wound healing and cell division [18,25]. Thus, the imbalance of zinc ions and its compounds and dysfunction of zinc transporters may result in many chronic diseases, including cancers, especially PCa [26,27,28,29]. In the past decades, substantial evidence has indicated low zinc levels in PCa samples and its relevance to tumor progression [30,31,32,33,34,35,36,37,38,39]. Importantly, many studies suggest that reduced zinc presence can serve as a potential biomarker for early diagnosis of PCa and may also be an intervening target for both consolidated and innovative PCa therapies [40,41,42,43]. The aim of this review is to summarize the evidence in the most recent literature with a focus on the levels, biological activities and relevant compounds of zinc in PCa. Moreover, this review also deliberates about the potential of clinical applications to target zinc signaling in PCa therapies.

## 2. Zinc Levels in Prostate Tissues and Sera of PCa Patients

Zinc is one of the most important microelements and ubiquitously present in the human body. Among soft tissues in our body, the prostate gland accumulates the highest levels of zinc, which is characteristic of healthy prostate [44,45]. The accumulation of cellular zinc leads to its increased levels in mitochondria, which inhibit m-aconitase activity and consequently prevent oxidation of citrate to isocitrate in the Krebs cycle [35,44,46]. This inhibition essentially truncates the Krebs cycle and diminishes the major cellular energy supply for excess proliferation and transformation, which can prevent normal prostatic epithelial cells (PrECs) from malignant transformation [35,44]. As a result, citrate accumulates at high levels in the prostatic reservoir and is secreted into prostatic fluid, where it plays an essential role in sperm motility and release [44]. Hence, zinc is critical for maintaining prostate health and inhibiting PCa development. Recently, evidence from several studies documented that aberrant zinc metabolism was closely related to the development of various types of malignant tumors, including breast, pancreas, lung, liver, stomach, cervix uteri and prostate cancers [26,28,47,48,49,50]. As early as 1950s, Mawson and Fischer first discovered that carcinomatous prostate tissues contained less zinc than normal prostate, and zinc content of any given prostate was directly associated with the proportion of alveolar tissue present [51]. Consistently, a number of reports in the following decades demonstrated that zinc levels were 62–85% lower in malignant prostate tissues than normal prostates using different zinc quantitative approaches [30,35,38,39,45,52,53,54,55]. Importantly, using in situ zinc staining methods to analyze human prostate tissue sections, researchers observed decreased zinc levels in malignant cells compared to normal PrECs [35,39,56,57]. Consistent with those observations, two additional studies using PCa cell lines revealed that LNCaP cells with relatively low malignancy contained a remarkably high cellular zinc concentration of 630 ng/mg proteins, as compared to the highly aggressive PC-3 cells of 191 ng/mg proteins [23,52]. Other reports also demonstrated a decreased trend of zinc levels during cancer progression and the association of low zinc levels with severe grades and advanced progression of PCa [30,39,58]. These data strongly indicated that severe loss of the zinc reservoir in diseased prostate tissues could serve as a hallmark of PCa development and progression.

Due to ease of access, zinc quantitation in sera is much easier than that in prostate tissue, the latter of which would cause prostatic damage. Therefore, it is necessary to investigate the relationship between serum and prostate zinc levels. Up to now, controversies still exist regarding the alteration of serum zinc presence in PCa patients. To date, eleven studies interrogating the correlation between serum and PCa zinc levels have been reported (Table 1). Among them, seven studies showed relatively low serum zinc levels in PCa patients versus normal individuals using atomic absorption spectrophotometry (AAS) assay [30,31,32,34,59,60,61], whereas two studies reported no association [62,63]. However, two of these studies claimed higher serum zinc concentrations in PCa patients than healthy individuals [64,65]. The inconsistencies among these reports could be due to sample sizes, measurement variations, setting of standards, sample collection, and difference in patient populations. In these reports, the classification of a PCa patient’s serum zinc level as “higher” or “lower” was based on whether the value is larger or smaller than the controls or the average serum zinc levels of healthy males; however, the values of these controls markedly varied among these studies (Table 1), likely due to the difference of employed zinc standards. Nevertheless, despite modest discrepancies, the available research data supported a reduced zinc presence in the sera of PCa patients. Certainly, further investigation is needed to precisely determine the correlation between serum zinc levels and PCa grades using a relatively large sample size and a well-defined zinc quantitation approach.

## 3. Biological Functions of Zinc in PCa

Reduced levels of zinc in malignant prostate tissues implicated its inhibitory role in PCa development and progression. Not surprisingly, a plethora of studies demonstrated tumor suppressive activities of zinc in the process of prostate tumorigenesis. At different stages of tumor development, the hallmarks of cancers comprise several biological characteristics, including sustained proliferation, resistance to cell death, induced angiogenesis and activated invasion and metastasis [66,67]. Hence, this section focuses on the effects of zinc on these features of PCa (Figure 1).

### 3.1. Zinc and Its Anti-Proliferative Activities

Maintaining cell proliferation is the first step of cancer development [66]. The first evidence showing the anti-proliferative effects of zinc in PCa dates back to 1999, when Liang et al. found that physiological levels of zinc could significantly inhibit LNCaP and PC-3 cell growth in a dose-dependent manner [68]. In this report, zinc could induce G2/M phase arrest through increasing p21 transcription in both LNCaP (wild type p53) and PC-3 (p53 null) cell lines [68]. Since then, a number of studies employed in vivo and in vitro approaches to demonstrate zinc-mediated repression of PCa cell proliferation through altering growth-related genes and signaling pathways [69,70,71,72,73,74,75,76,77], which reinforced the anti-proliferative activity of zinc in PCa.

Mitogen-activated protein kinases (MAPKs) regulate cell proliferation through pathways involved of ERK1/2, JNK and p38 activation [78,79]. In 2008, Wong and Abubakar showed that increased intracellular zinc levels activated ERK1/2 phosphorylation through the VHR/ZAP-70-associated pathway, which eventually repressed proliferation of LNCaP cells [72]. In 2009, Han et al. reported that zinc deficiency could promote cell proliferation of both normal and malignant human prostate cells but through distinct mechanisms [73]. In human normal PrECs, zinc deficiency could enhance cell growth through the PTEN/AKT/MDM2/p53 signaling axis. Mechanistically, low levels of zinc inactivated the tumor suppressor PTEN through promoting its phosphorylation, leading to AKT hyperphosphorylation and activation. Activated AKT could phosphorylate MDM2 to promote its nuclear retention, which would facilitate p53 ubiquitination and degradation, consequently leading to increased cell survival [73]. However, in LNCaP cells deficient of active PTEN, lack of zinc could induce the AKT/p21 signaling axis to promote cell growth [73]. In this signaling axis, zinc deficiency led to AKT hyperphosphorylation and activation, which promoted p21 phosphorylation and consequently blocked its nuclear entry, causing p21 cytoplasmic retention and degradation. Thus, loss of p21 would enhance LNCaP cell proliferation through promoting G0/G1 transition in the cell cycle [73].

A recent study indicated that zinc could sensitize PC-3 cells to chemotherapeutic treatment by paclitaxel, a generic anticancer medicine [74]. In addition, the treatment of androgen receptor (AR)-retaining PCa cells by zinc chloride could significantly suppress cell proliferation through decreasing AR expression [75]. Most recently, Hacioglu et al. demonstrated an anti-proliferative effect of zinc sulphate on human PCa DU-145 cells in a dose-dependent manner [76]. Interestingly, a study by Wong et al. indicated that continuous exposure to supraphysiologic concentration of Zn could increase the expression of cancer promoting genes FBL and CD164 in LNCaP cells [77]. Overall, the majority of in vitro studies suggested that zinc at appropriate or physiological levels could inhibit PCa cell proliferation through modulating different genes and signaling pathways.

Consistently, in vivo studies also indicated anti-proliferative activities of zinc in PCa [80,81,82,83,84]. Among these reports, Shah et al. found that direct intratumoral injection of zinc acetate to PCa tumors of a xenograft model could repress tumor growth and substantially extend the survival of the mice [81]. Importantly, this treatment did not cause any detectable cytotoxicity in other tissues or organs, including brain, heart, kidney and liver. Later on, the same group investigated the effect of zinc supplementation on PCa development in the transgenic adenocarcinoma of the mouse prostate (TRAMP) model. In this study, the authors observed that both zinc deficiency and higher levels of dietary zinc could increase tumor growth, whereas diet with optimal zinc levels could attenuate tumor development. The normal zinc diet could decrease the expression of the proliferative gene IGF-1 and reduced the ratio of IGF-1/IGFBP-3 in the TRAMP mice [82]. These results suggested that an optimal dietary zinc intake would play a protective role against PCa through modulating the IGF-1 signaling. Recently, Gu et al. showed that a combined treatment of Pmp53, a plasmid containing both MDM2 small hairpin RNA and the wild type p53 gene, with zinc could markedly inhibit tumor growth through increasing p21 expression and inducing G1 arrest in tumor xenografts of PC-3 cells, as compared to the treatment group with only Pmp53 [83]. Fong et al. observed that nuclear staining of a proliferation marker, PCNA, was obviously increased in zinc-deficient prostates compared to zinc-sufficient prostates in middle-aged male rats [84]. Their data suggested that dietary zinc deficiency could promote prostate epithelial cell proliferation, which is in agreement with the above perspectives.

In summary, research data from different groups strongly indicates that zinc could play a suppressive role in PCa cell proliferation through regulating various gene expression or activities and mediating intra- and intercellular signaling pathways.

### 3.2. Zinc and Its Role in Cell Death

Resistance to cell death is an important characteristic of cancer cells [66]. In normal prostate tissue, zinc can attenuate the Krebs cycle to reduce ATP production [42,44]. Furthermore, high zinc levels promote the release of cytochrome c from mitochondria into cytosol through facilitating the insertion or docking of BAX and its subsequent oligomerization to form pores in the mitochondrial membrane (Figure 1). As a result, cytochrome c released into cytosol activates the caspase cascade to promote apoptotic cell death [44,85]. Thus, excess zinc may induce cell death through directly activating BAX-mediated mitochondrial apoptosis in normal PrECs. Indeed, a line of studies demonstrated that zinc treatment could induce apoptosis through loss of mitochondrial membrane integrity, release of cytochrome c, increasing levels of BAX and/or BAX/BCL2 ratio and activation of caspase-9 and caspase-3 in a variety of PCa cell lines, including PC-3, HPR-1, 22Rv1, LNCaP and DU145 [69,70,76,86,87,88]. Importantly, Feng et al. employed an in vivo nude mouse model to demonstrate that zinc treatment could repress tumor growth through increasing the BAX/BCL2 ratio to induce apoptosis [89]. Additionally, zinc also modulates other apoptosis-related genes and pathways to induce cell death in PCa cells. For example, zinc was shown to significantly decrease expression of the anti-apoptotic or pro-survival genes BCL2 and survivin, leading to PCa cell apoptosis [90]. Yang et al. reported a new mechanism of zinc-induced apoptosis, in which zinc could recruit the Smad2/4/PIAS1 complex to promote p21 gene expression, leading to PCa cell death [91]. In this model, zinc treatment increased the expression of Smad2 and PIAS1, which associated with Smad4 to form the Smad2/4/PIAS1 complex. This complex was subsequently translocated into the nucleus and bound to the p21 promoter through zinc recruitment, leading to activated p21 expression and accelerated apoptosis [91]. Overall, these in vitro and in vivo studies strongly suggested a pro-apoptotic activity of zinc through regulating different apoptosis pathways in PCa cells.

In addition to zinc-induced apoptosis in PCa cells, Carraway and Dobner demonstrated that ionophore Zn-pyrithione (ZP) could induce a strong necrotic response through activating ERK1/2 and protein kinase C (PKC) in PCa cells [92]. Consistently, another group also reported that zinc sulfate treatment could promote necrosis of LNCaP and PC-3 cells in a time- and dose-dependent manner [90].

Besides well-studied apoptosis, several other types of regulated cell death can also cause cell decease, including necroptosis, pyroptosis, ferroptosis, phagoptosis and entosis, while autophagy can lead to either cell death or survival [93,94,95,96]. Zinc-mediated PCa cell death through these additional mechanisms has not been reported to date, although it was shown in many other cancers [97,98,99]. However, therapies targeting these cell death types have become a new and effective strategy to treat and prevent PCa in the clinic [93]. Therefore, future efforts are needed to investigate whether zinc can directly or indirectly regulate these cell death mechanisms during PCa development and progression, which may provide insights into designing novel therapeutic strategies.

### 3.3. Zinc and Its Anti-Metastasis Effects

Metastasis is the major cause of death in PCa patients [66]. The characteristics of canonical cancer metastasis include epithelial-to-mesenchymal transformation (EMT), migration, invasion, extracellular matrix (ECM) interactions and angiogenesis [66,67]. First, the EMT is an early step of metastasis, in which epithelial cells lose cell–cell adhesion and acquire a spindle-like morphology with migratory and invasive capabilities [66,67]. As an intercellular adhesion molecule, intercellular adhesion molecule (ICAM-1) plays an important role in cell–cell and cell–extracellular matrix interactions in the process of tumor invasion [100,101]. An in vitro study by Uzzo et al. demonstrated that zinc sulphate treatment in PC-3 cells could reduce the expression of ICAM-1, as well as other angiogenic and metastatic factors, including vascular endothelial growth factor (VEGF), IL-8 and MMP-9 [71]. Aminopeptidase N (AP-N) can degrade collagen type IV to promote cell motility and adhesion to the extracellular matrix; therefore, it plays a critical role in tumor invasion [66,102,103]. Ishii et al. observed that AP-N extracted from human prostate could be irreversibly repressed by low concentrations of zinc [104]. The results from these two studies indicated that zinc could suppress EMT through inhibiting adhesion molecule ICAM-1 expression and AP-N activity.

Second, following the EMT, cancer cells need to acquire migratory and invasive capabilities [67]. Zinc could inhibit PCa cell migration and invasion. In a study using transwell cell culture chambers, high zinc concentration with bestatin could significantly suppress PC-3 cell invasion into Matrigel [104]. Another study from the same group demonstrated that zinc ions remarkably inhibited invasion of LNCaP cells [105]. Their study also revealed that zinc could inhibit the activity of PSA in cleaving ECM components and thus block its function in promoting PCa metastasis. Thus, their finding suggested that zinc could suppress PCa cell migration and invasion through inhibiting PSA activity.

Third, the ECM reconstruction, tumor cell migration and invasion are accompanied by angiogenesis, which ultimately enables metastasis [66,67,106]. Angiogenesis is regulated by angiogenic stimulating factors, such as angiogenin, VEGF, basic fibroblast growth factor (bFGF), transforming growth factor β (TGFβ), cytokines (IL-1, 6 and 8) and many other inflammatory factors [66,106,107]. The anti-angiogenic properties of zinc in PCa were previously reported by several groups. Uzzo et al. demonstrated that zinc treatment could inhibit angiogenic activity in PCa cells, predominantly through modulating MAPKs, NF-κB and several angiogenic factors, including VEGF, IL-6, IL-8 and MMP-9 [71]. Mechanistically, zinc supplement promoted phosphorylation of the members of three major MAPK subfamilies: ERK1/2, JNK and p38. Zinc could also inhibit TNF-α-activated NF-κB through stabilizing the inhibitory subunit IκBα, leading retarded nuclear translocation of RelA. The inhibition of NF-κB could transcriptionally repress the expression of pro-angiogenic factors, such as VEGF, IL-6, IL-8 and MMP-9, compromising the angiogenic potential of PCa cells [71]. Similarly, Golovine et al. reported that zinc at physiological levels could inhibit NF-κB activity in androgen-independent PC-3 and DU-145 cells, and decrease the expression of multiple prognostic marks for poor clinical outcomes, including VEGF, IL-6, IL-8, and MMP-9 [108]. The treatment with the zinc chelator *N*,*N*,*N*′,*N*′-tetrakis(2-pyridylmethyl)-ethylenediamine (TPEN), a specific agent to induce zinc deficiency, could reverse these effects [108]. The data suggested that zinc loss could promote tumor progression through activating NF-κB and its downstream oncogenic pathways.

Hypoxia inducible factor-1α (HIF-1α) plays a major role in HIF-1-induced gene expression under hypoxic conditions and undertakes an “angiogenic switch” function during tumor metastasis [109,110]. In human PCa C38 and C27 cells, zinc could induce proteasomal degradation of HIF-1α and consequently reduce HIF-1α recruitment to the VEGF promoter and its transactivation of this target gene, leading to decreased invasiveness and tumor formation [109]. These data suggested that zinc could inhibit PCa angiogenesis through VEGF-regulated pathways.

It is well appreciated that tumor microenvironment (TME) plays a critical role in promoting metastasis [111,112,113]. TME consists of malignant cells and a wide range of surrounding non-cancer cells, including cancer-associated fibroblasts, tumor-infiltrating immune cells, secreted soluble factors (cytokines, chemokines and exosomes) and non-cellular support materials (such as extracellular matrix, or ECM) [114,115]. As we reviewed above, zinc inhibits the expression of ICAM-1 and AP-N and reduces the production of various angiogenesis stimulating factors and inflammatory factors (VEGF, IL-6, IL-8 and MMP-9). In addition, hypoxia is a major characteristic of TME, which promotes angiogenesis, lymphangiogenesis and inflammation, leading to nutrient supplement and enhanced metastasis by inflammatory cells [116,117]. Zinc could decrease VEGF expression via promoting proteasomal degradation of HIF-1α in a hypoxic condition, which in turn reduced PCa cell invasion and progression [109]. Therefore, zinc can damage tumor vascular microenvironment through inhibiting the release of cytokines and express adhesion molecules within TME to reduce PCa metastasis. However, no evidence supports any direct effect of zinc on surrounding non-tumorigenic cells, such as immune cells, within the microenvironment of PCa. Noticeably, as we reviewed above, zinc exerts an inhibitory activity on NF-κB pathway [71,108]. Previous studies indicate that NF-κB regulates immune functions within the TME and promotes the function of the CD4+ regulatory T (Treg) cells to inhibit antitumor immune response [118,119]. Thus, zinc’s negative regulation of the NF-κB pathway suggests its role in activating immune cells within the TME to attenuate PCa development. Consistently, in the studies of leukemia and head and neck cancer, zinc was reportedly to regulate the number and function of tumor-associated host immune cells, such as T helper 1 (Th1), Th2, Th17, Treg, dendritic cells, macrophages and myeloid-derived suppressor cells, which are major components of the TME [118,120,121]. In addition, multiple studies demonstrated a regulatory role of zinc in TME-related immune response [18,118,122]. Despite these observations, whether reduced zinc levels directly impact on tumor-associated host immune cells within the TME of PCa needs future investigation.

Taken together, zinc can act as a signaling factor to diminish multiple metastatic characteristics of PC cells, including EMT, migration, invasion, ECM interactions and angiogenesis, through various pathways.

## 4. Zinc-Associated Compounds and Their Functions in PCa

The aforementioned evidence indicated a tumor suppressive role of zinc in PCa development and progression. This also suggested that proteins either regulating zinc homeostasis or using it as a functional ligand may also play vital roles in prostate malignant transformation. In this section, we will discuss the expression and functions of important zinc compounds in PCa, including zinc transporters and zinc finger transcription factors.

### 4.1. Zinc Transporters

Zinc homeostasis in the prostate gland is tightly regulated by two protein families of zinc transporters; the ZIP family mediates zinc transport from extracellular fluid or intracellular vesicles into cytoplasm, and the ZnT family is responsible for decreasing cytoplasmic zinc levels by transporting zinc from cytoplasm to the outside of cells or into intracellular vesicles [123,124,125]. To date, accumulating evidence suggests that aberrant expression of ZIP and ZnT family proteins is closely related to PCa development and progression [23,44,85]. The ZIP family has 14 members in humans, but only ZIP1-8 and 14 have been shown to possess zinc transporting activity [24,125,126,127]. A number of reports and reviews demonstrated prominent downregulation of ZIP1-4 in PCa as compared to normal prostate or benign prostate hyperplasia (BPH) tissues [23,28,35,39,56,84,128,129]. Although all four of these ZIP proteins were localized in the plasma membrane of prostate epithelia and involved in importing zinc from extracellular fluid, ZIP1 was present at the basolateral membrane and played the most important role in zinc uptake [39,126,130]. Consistently, Franklin et al. observed that ectopic ZIP1 expression remarkedly increased zinc uptake and accumulation in PC-3 cells, leading to retarded cell proliferation. Meanwhile, ZIP1 silencing by its antisense oligonucleotide could significantly decrease zinc uptake [130]. The following two studies reinforced the importance of ZIP1 in PCa. Zhang et al. observed that ZIP1 introduction into nontumorigenic prostate RWPE2 cells could elevate intracellular zinc levels and suppress cell growth through promoting apoptosis [131]. In the report of Golovine et al., ZIP1 overexpression could reduce the tumorigenic potential and growth of PCa cells through inhibiting the expression of NF-κB-dependent angiogenic and pro-metastatic cytokines both in vitro and in vivo [132]. In addition, several groups explored the mechanisms underlying ZIP1 downregulation in PCa. Zou et al. reported that overexpression of the Ras-responsive element-binding protein 1 (RREB1) transcription factor could downregulate ZIP1, leading to low zinc presence in PCa [23,133]. Mihelich et al. showed that the microRNA-183-96-182 cluster could target the 3′-UTR of the ZIP1 mRNA to inhibit its expression in PrECs and thus block zinc uptake [23,134]. These results indicate that ZIP1, a major zinc uptake transporter, functions as a tumor suppressor in PCa, which is consistent with the anti-tumor activity of zinc discussed above.

In contrast to ZIP1 localization at basolateral membrane, ZIP2 and ZIP3 are predominantly distributed at the apical cell membrane of prostate epithelial cells [39,44,128]. Thus, ZIP1 is crucial for zinc extraction from circulation (the primary source of cellular zinc), while ZIP2 and ZIP3 are apparently associated with zinc retention in cellular compartments [23,39,128]. ZIP4 is also localized at the apical membrane [44,129], suggesting its role is similar to ZIP2 and ZIP3 in mediating zinc homeostasis. Exogenous ZIP4 expression exerted inhibitory effects on the proliferation and invasiveness of DU145 cells, and its silencing exhibited the opposite impacts on 22RV1 cells [129], suggesting its tumor suppressive activity in PCa development. One study demonstrated a statistically significant downregulation of ZIP14 in the prostate of zinc-deficient middle-aged rats, resembling the profiles of human PCa, but the activity of ZIP14 in zinc homeostasis was not explored [84]. To the best of our knowledge, the expression and functional roles of other ZIP proteins in PCa have not been reported.

Taken together, these data collectively provided compelling evidence showing that reduced cytoplasmic zinc levels in PCa cells mainly depend on downregulation of ZIP1 and the absence of other zinc transporters with maintenance functions, including ZIP2, ZIP3 and ZIP4.

Compared with ZIP transporters, only scarce information about the members of the ZnT family is available in PCa-related studies. To date, 10 members of the ZnT family have been identified in humans, but only ZnT1 was shown to localize to the plasma membrane [26,125,126,135]. Different studies indicated that ZnT1 mRNA levels were increased [84,126], decreased [43,136] or unchanged [137] in PCa samples, as compared to BPH tissues. Likewise, several groups reported that ZnT4 and ZnT5 expression levels were indeterminant between PCa samples and BPH tissues [44,137,138]. Additionally, ZnT2, ZnT3 and ZnT8 were reportedly undetectable in human PCa tissue samples [84]. As compared to BPH tissues, PCa samples showed significantly reduced ZnT6 expression, but their ZnT9 and ZnT10 levels were markedly elevated [44,84,126]. In this comparison, ZnT7 expression was either increased or unchanged in PCa samples [44,84]. Importantly, in a study using the TRAMP mice, ZnT7-null status could promote PCa development through decreasing cell apoptotic potential [139]. Mechanistically, ZnT7 played a role in the early secretory pathway, primarily on the cis-face of the Golgi apparatus. Thus, the ZnT7 null condition could block zinc transport from the cytoplasm into the Golgi apparatus, causing accelerated PCa development, which is seemingly not in accordance with the anti-growth activity of cytoplasmic zinc. The results of this study implicated an essential role of zinc presence or enrichment in specific subcellular compartments to PCa progression. More importantly, the coordinated zinc mobilization by ZIP and ZnT transporters is a critical requisite for maintaining zinc homeostasis via regulating its influx and efflux in mammalian cells (Figure 1). Therefore, future exploration is needed to delineate the mechanisms underlying the cooperation between the members of the ZIP and ZnT families to maintain zinc homeostasis and its dysregulation during PCa development. Whether different zinc transporters of the two families have any functional redundancy or expression compensation in prostate cells also deserves further investigation.

### 4.2. Zinc Finger-Containing Transcription Factors

Deregulated levels and activities of TFs can alter the expression of cancer-related genes and consequently contribute to tumorigenesis [140,141,142,143]. With accurately regulated homeostasis by ZIPs and ZnTs in cells, zinc undertakes various biological functions through modulating the structures and functions of ZF-containing TFs. Zinc is a regulatory component of ZF proteins, which include a large number of TFs [144].

Based on the numbers of cysteine and/or histidine residues involved in zinc ion bindings and other structural features, ZFs can be stratified into different classes, including ZnHis_2_Cys_2_ (C_2_H_2_), ZnHisCys_3_ (C_2_HC), ZnCys_4_ (C_2_C_2_, C_4_) and Zn_2_Cys_6_ (C_2_C_2_C_2_, C_6_) fingers [144,145]. Among them, the C_2_H_2_ is a classical and the most common type, which is a self-contained domain stabilized by a zinc ion bound to a pair of cysteines and a pair of histidines, as well as by an inner hydrophobic core structure [145,146]. In addition, based on the number, localization and arrangement of ZFs within a molecule, the C_2_H_2_-type ZFs can be divided into 4 subclasses, including single-fingered, triple-fingered, separated-paired-fingered and multiple adjacent-fingered proteins [144,147]. Furthermore, C_2_H_2_-type ZF proteins also contain other structural domains, including the Broad-complex, Tramtrack, Bric-a-brac (BTB)/poxvirus and zinc finger (POZ), Krüppel-associated box (KRAB) and SCAN domains. Most of these domains are positioned at the N-termini and function as platforms for protein-protein interactions, while ZFs are generally located at C-terminal regions [144,148]. Noticeably, genotoxic or cytotoxic challenges, such as oxidative stress, nitrogen oxide and DNA mutagenesis, could impair the binding force of zinc, leading to functional loss of ZFs, especially during tumorigenesis [149,150,151]. Hence, ZFs can serve as potential intervening targets in cancer therapies.

Aberrant expression and mutations of ZF-containing TFs may contribute to PCa initiation and development, suggesting an essential role of zinc in intracellular signaling. Here, we use three examples to discuss the functional activities of ZF-containing TFs, including AR, specificity protein 1 (SP1) and promyelocytic leukemia zinc finger (PLZF). Currently, these three TFs or their related signaling pathways are considered as potential targeting points in PCa therapies.

AR is a representative member of the C_4_-type ZF class, which is critically important to PCa development and progression [152,153]. The AR protein has four distinct functional domains from its N- to C-end: an N-terminal transactivation domain (NTD), a DNA binding domain (DBD) containing two highly conserved C_4_-type zinc fingers, a flexible hinge region, and a C-terminal ligand binding domain (LBD) [154,155]. AR is highly expressed in both primary and metastatic PCa with either hormone sensitive or refractory characteristics [156,157,158]. Persistent activation of the AR pathway is mainly dependent on its gene amplification and mutations, including generation of truncated AR splicing variants lacking the LBD, such as AR-V7 and AR-V12 [159,160,161,162,163]. In the clinic, high AR levels correlated with short recurrence-free survival and disease progression [157,164]. Mechanistically, androgen molecules, including testosterone and dihydrotestosterone (DHT), can bind the LBD to activate AR, which then primarily associates with its responsive binding elements on the target promoter androgen-responsive genes and promotes their expression, leading to PCa growth and metastasis [160,165]. A missense mutation caused by a single base replacement in the ZF locus encoding its DNA-binding domain could lead to AR dysfunction [166], indicating the essential role of the DBD for AR activities.

SP1 is a representative protein of the C_2_H_2_-type ZF class and involved in prostate tumorigenesis. As an increasingly expressed protein during prostate malignancy, SP1 could potentially serve as a prognostic marker in PCa patients [167]. Based on its protein structure, SP1 is a triple-fingered ZF protein containing a highly conserved DNA-binding domain composed of three zinc fingers at the C-terminal [168,169]. As a well-characterized TF, SP1 regulates the expression of many important genes, including AR, TGF-β, c-Met, fatty acid synthase (FASN), matrix metalloprotein (MT1-MMP), PSA and α-integrin through binding to its consensus sites in their promoters [167,170,171,172].

Another classical C_2_H_2_-type ZF, PLZF, is featured with multiple adjacent-fingered ZF and BTB/POZ domains [173,174]. Different from AR and SP1 described above, PLZF consists of a BTB/POZ domain with repressive function, a second repressive domain (RD2), and a zinc finger domain consisting of nine Krüppel-like C_2_H_2_ ZFs responsible for binding to its target promoters. The BTB/POZ domain allows PLZF to form either homodimers or heteromeric repressive complexes with NCoR, Sin3A and HDAC1 [175,176,177,178]. The PLZF gene is generally downregulated or lost in high grades of PCa samples as compared to low grade ones [179,180]. PLZF could block PCa cell growth through its inhibition to AR, mTOR and MAPK signaling [181,182]. Furthermore, in vitro and in vivo data from different groups revealed that loss of PLZF promoted PCa cell growth [181,182,183,184], suggesting its tumor suppressive activity.

Overall, zinc is an essential structural component in ZF-containing TFs and required for their DNA binding activities. Therefore, altered zinc levels in blood and prostate glands are closed related to dysregulated function of ZF-containing proteins.

## 5. Clinical Applications of Zinc Signaling in PCa

Significantly decreased zinc levels during prostate malignancy implicated its activities in inhibiting proliferation and metastasis of tumor cells and inducing cell death, which led to the development of zinc or its related compounds in diagnostic and therapeutic applications of PCa. Due to the controversies and inconsistent results regarding the effects of zinc supplementation on PCa among different laboratory research and epidemiologic studies [23,43,80,82,89,185,186,187], we will mainly discuss the applications of zinc and its associated proteins as they relate to clinical diagnosis of PCa in this section. Additionally, we will comment on immunotherapies targeting zinc signaling.

In the clinic, currently prevailing tests of PCa diagnosis can be divided into two categories: traditional and modern methods. The traditional methods include digital rectal examination and blood PSA tests, while modern methods embrace targeted magnetic resonance imaging (MRI), ultrasound fusion prostate biopsy and conventional radiological imaging [188,189,190]. Each approach may have its own disadvantages in specificity, invasiveness or targeting accuracy, which restricts its applications to patients with specific types or stages of the disease [41,58]. Fortunately, PCa is the only known prostatic disease associated with a substantial decrease of zinc levels [45]; neither prostatitis nor BPH exhibit this phenomenon [41,191], suggesting that zinc serves as an excellent candidate biomarker for PCa. Indeed, based on synthetic images generated from clinical data of zinc distributions, zinc-based diagnostics could represent an approach superior to the serum PSA test [58,192]. Recently, several groups developed in vivo imaging strategies to simultaneously probe zinc presence and detect PCa progression [40,41,58,193,194]. Ghosh et al. employed a novel fluorescent zinc sensor ZPP1 that could precisely bind two zinc ions to monitor cell malignant transformation in the TRAMP model and observed tumor progression related to decreasing fluorescence intensity in an age-dependent manner. This study is the first report of using altered zinc levels as an innate imaging biomarker for early PCa detection [58,195]. Due to the limitations of optical imaging [41], several groups attempted to optimize zinc measurement using MRI in the following years. Jordan et al. discovered a zinc-binding gadolinium using a paramagnetic contrast agent and used it to detect extracellular zinc by proton MRI following glucose-stimulated zinc secretion. This strategy let them differentiate healthy versus malignant mouse prostates, which could provide a novel and highly specific approach for PCa diagnosis [40]. More recently, using MRI based on 19F ion chemical exchange saturation transfer (iCEST) and TF-BAPTA as a fluorinated zinc probe, Yuan et al. was able to discriminate normal and malignant prostate cells with a 10-fold higher sensitivity than the method based on glucose-stimulated zinc secretion. The iCEST-MRI allowed them to observe over 300% gradual zinc decrease in the in vivo transition of normal PrECs to cancer cells [41]. This study is the first attempt to use the 19F iCEST-MRI as a diagnostic tool for in vivo zinc imaging. Since both iCEST and 19F MRI are clinically used, this approach possesses high translational potential for clinical diagnosis of PCa. Despite these promising research and preclinical data, further exploration needs to focus on developing zinc detection strategies with high specificity, sensitivity, and economic advantage to achieve early PCa diagnosis.

Noteworthily, decreased intraprostatic zinc levels generally coincide with significantly reduced expression of the zinc transporters ZIP1, ZIP2, ZIP3 and ZIP4, which represents an early step in PCa development [23,39,41,128,129]. Based on the impacts of altered expression of these zinc transporters on PCa cell growth and metastasis, the expression levels of ZIP1, ZIP2, ZIP3 and ZIP4 genes may also serve as potential biomarkers for early PCa diagnosis. Additionally, among the upstream regulators of ZIP1 (RREB-1 and microRNA-183-96-182), the proteins modulating key zinc signaling pathways (NF-κB, PI3K and MAPK), and ZF-containing TFs (AR, PLZF and SP1), many of them have been evaluated as or determined to be potential assistant biomarkers for PCa diagnosis. In our opinion, zinc status and the genes involved in zinc homeostasis could serve as an adjunctive measure to the traditional and modern methods of PCa diagnosis.

In the past decade, immunotherapy has proven to be an effective approach in the treatment of multiple cancer types, especially melanoma and non-small cell lung cancer [196,197,198]. For PCa, immunotherapies using immune checkpoint inhibition, PSA vaccines and dendritic cell-based strategies have been intensively tested in clinical trials [199]. Ample evidence demonstrated zinc’s contribution to the maintenance of host systemic immune system, and thus, its moderate levels could decrease inflammation and oxidative stress [200,201,202,203]. Generally, zinc at its physiological levels is essential to the growth, differentiation and biological function of various immune cells, including macrophages, dendritic cells, neutrophils, mast cells, T cells and B cells [204,205,206,207,208,209]. On the other hand, zinc deficiency leads to impaired immune response and an increased risk of inflammation and tumorigenesis [201,205,210]. Consistently, moderate zinc supplementation can restore or even improve host defense and reduce both morbidity and mortality of various diseases, including cancers [211,212,213]. Therefore, targeting zinc signaling to prevent immune escape of tumor cells and promote immune cells to eradicate cancers represents a logical and promising strategy in the treatments of PCa patients. However, due to the high complexity of the immune microenvironment and high heterogeneity of antitumor immune responses [199,214,215], the application of targeting zinc signaling in immunotherapies has not been tested in either preclinical models or the patients of PCa.

## 6. Conclusions and Future Prospects

Zinc is a vital nutrient element for human health. Interestingly, the prostate is known to host the highest levels of mobile zinc among all soft tissues of the human body. The high zinc content in prostatic fluid is required for optimal male fertility due to its antimicrobial activity and ability to sustain sperm motility. Major and ample evidence consistently supports the notion that zinc levels are markedly reduced in PCa. In addition, multiple in vitro and in vivo studies demonstrated antitumor activities of zinc in PCa, including suppression of cell proliferation, induction of cell apoptosis and inhibition of metastatic processes, including EMT, migration, invasion, ECM interactions and angiogenesis (Figure 1). All these data strongly suggest that zinc is an intra- and intercellular signaling mediator and can serve as a bona fide tumor suppressor factor during prostate tumorigenesis. Due to the importance of zinc homeostasis to prostate health, zinc transporters, including ZIP1, ZIP2, ZIP3 and ZIP4, also exhibit tumor suppressive functions in PCa development and progression. Furthermore, zinc modulates the transcriptional activity of ZF-containing TFs, such as AR, PLZF and SP1, which regulate different signaling pathways in PCa. Finally, accumulating research data strongly suggest mobile zinc as an excellent candidate biomarker for clinical diagnose of PCa. More importantly, adding the analysis of a zinc signaling gene expression profile to traditional and modern clinical detection methods represents an effective and reliable strategy to more accurately diagnose PCa patients at early stages. Despite the achievements summarized above, we propose the following perspectives of zinc studies in PCa in future. First, we need to precisely and firmly determine the association between serum zinc levels and its content in PCa. Second, we should delineate the pathways or molecular mechanisms underlying zinc-mediated cell death, especially ferroptosis in PCa development and progression. Third, the expression status and biological activities of currently unexplored ZIPs and ZnTs in PCa need to be clarified. Fourth, whether ZIPs and ZnTs cooperatively regulate zinc homeostasis in healthy prostate and PCa deserves further investigation. Fifth, the preventative activity of zinc supplementation against PCa requires extensive exploration using a large population. Sixth, development of a specific, sensitive and economic approach to detect zinc levels in prostate tissues for early PCa diagnosis is urgently needed. Seventh, targeting zinc signaling in PCa immunotherapies needs to be verified using preclinical models.

## Figures and Tables

**Figure 1 ijms-21-00667-f001:**
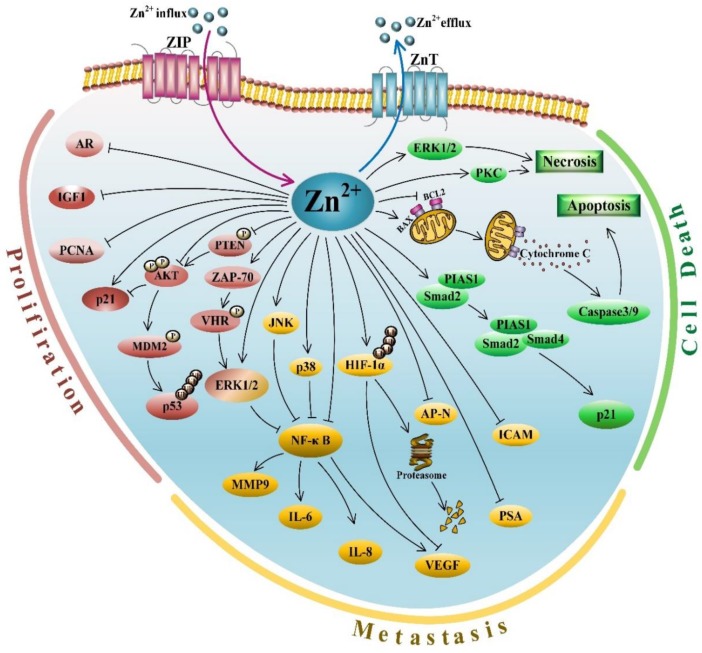
An overview of molecular signaling pathways mediated by zinc in PCa. Zinc is involved in various antiproliferative and proapoptotic pathways to exert its antitumor activities, including suppressing cell proliferation, inducing cell death and inhibiting metastasis. Zinc-regulated molecules involved in proliferation, metastasis and cell death, as well as zinc transporters, are grouped and differentially colored. Black T bar: inhibiting function of zinc or proteins; Blue arrow: transporting zinc from cytoplasm to extracellular fluid; Purple arrow: transporting zinc from extracellular fluid into cytoplasm; Black arrow: activating function of zinc or proteins.

**Table 1 ijms-21-00667-t001:** Summary of reports comparing serum zinc levels between PCa patients and healthy controls.

PCa/Control (Serum Zinc Level, µg/dL)	PCa/Control (Number)	Testing Assay	Cohort Sources/Populations	Age Medians or Ranges (PCa/Control)	References
61.60 ± 19.75/ 99.59 ± 29.23	220/220	AAS	Nigeria	69.73 ± 6.32/ 68.97 ± 5.76	Wakwe et al. 2019 [30]
576 ± 102/ 711 ± 164	25/24	AAS	Turkey	67.5 ± 8.8/ 65.0 ± 6.0	Aydin et al. 2006 [31]
63.40 ± 6.40/ 86.50 ± 15.20	18/20	AAS	India	55−85/ 30−50	Christudoss et al. 2011 [32]
4.66 ± 2.22/ 19.26 ± 3.26	30/32	AAS	Turkey	65.4 ± 4.2/ 62.8 ± 5.8	Kaba et al. 2014 [34]
8300 ± 213/ 9780 ± 257	42/101	AAS	China	70.1 ± 1.32/ 67.8 ± 0.85	Li et al. 2005 [59]
147.75 ± 42.05/ 168.78 ± 59.80	50/50	AAS	Nigeria	50−70/ 50−70	Onyema-iloh et al. 2015 [60]
63.92 ± 19.10/ 103.61 ± 32.43	85/90	AAS	China	64.7 ± 9.2/ 65.9 ± 8.4	Chen et al. 2015 [61]
91.55 ± 12.42/ 90.89 ± 12.42	50/10	AAS	German	68.6/65.9	Feustel et al. 1989 [62]
94.09 ± 20.40/ 93.9 ± 17.60	392/783	AAS	America	69.1 ± 7.1/ 68.9 ± 7.2	Park et al. 2013 [63]
89.89 ± 1.20/ 85.66 ± 1.31	197/197	AAS	Poland	72/72	Białkowska et al. 2018 [64]
112.93 ± 18.10/ 98.12 ± 8.24	40/28	AAS	China	ND/ND	Yao et al. 1977 [65]

AAS, atomic absorption spectrophotometry; ND, not described in studies.

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
