# Peer review of "Advances of Zinc Signaling Studies in Prostate Cancer"

_ijms, 2020, doi:10.3390/ijms21020667_

Round 1
Reviewer 1 Report
In this manuscript, authors review literature associated with various aspects of the zinc study in prostate cancer (PCa), including the level, signaling, chemicals and clinical application. Based on this review, zinc plays a tumor suppressive role and could be applied as a potential biomarker in PCa, and zinc-containing compounds could be used a therapy to treat this disease. Overall, this is a well-written review article and provides a comprehensive information regarding zinc research in PCa. Few questions need to be addressed before publication:
In Table 1, how to define “Lower or Higher” for serum zinc levels? Could authors showed the real numbers in Table 1, if possible. One of key issues for cancer progression is tumor microenvironment, how zinc impacts on tumor microenvironment during cancer aggressiveness? Cancer immunotherapies, including immune checkpoint therapy, cell therapy, cancer vaccine, and others, have been well reported and demonstrated to be one of the attractive and promising treatments to fight cancers. Any publication or evidence showed that zinc might be involved in or contributed to the host systemic immune system which could be associated with cancer Immunotherapies?Author Response
Response to Reviewer 1 Comments
Title: Advances of Zinc Signaling Studies in Prostate Cancer
Manuscript ID: ijms-679303
In this manuscript, authors review literature associated with various aspects of the zinc study in prostate cancer (PCa), including the level, signaling, chemicals and clinical application. Based on this review, zinc plays a tumor suppressive role and could be applied as a potential biomarker in PCa, and zinc-containing compounds could be used a therapy to treat this disease. Overall, this is a well-written review article and provides a comprehensive information regarding zinc research in PCa. Few questions need to be addressed before publication:
Point 1: In Table 1, how to define “Lower or Higher” for serum zinc levels? Could authors show the real numbers in Table 1, if possible.

Response 1: In each study, the “Lower” or “Higher” for the serum zinc level of a PCa patient was defined based on the value “lower” or “higher” than the serum zinc levels of the control group (healthy males) in each study. We have indicated this defining rule in the revised manuscript. As suggested by this reviewer, we included the real numbers of serum zinc levels (PCa/control) in Table 1 of the revised manuscript.
Point 2: One of key issues for cancer progression is tumor microenvironment, how zinc impacts on tumor microenvironment during cancer aggressiveness?
Response 2: We thank this reviewer for the important suggestion. Accordingly, we have added substantial discussion regarding the contribution of zinc to the tumor microenvironment of PCa in the Section 3.3 (Zinc and its Anti-Metastasis Effects). The paragraph of these discussions is on Page 8 of the revised manuscript.
Point 3: Cancer immunotherapies, including immune checkpoint therapy, cell therapy, cancer vaccine, and others, have been well reported and demonstrated to be one of the attractive and promising treatments to fight cancers. Any publication or evidence showed that zinc might be involved in or contributed to the host systemic immune system which could be associated with cancer Immunotherapies?
Response 3: We thank this reviewer for proposing this important point. Based on the suggestion, we have reviewed and commented the potential role of zinc in prostate cancer immunotherapies in the Section 5 (Clinical Applications of Zinc Signaling in PCa). The paragraph of these discussions is on Page 12 of the revised manuscript.
Reviewer 2 Report
The authors declare that the document I have reviewed is a review. It has not been described in any way how the search for sources was made. In the "Instruction fro Authors" you can read:
Review manuscripts should comprise the front matter, literature review sections and the back matter. The template file can also be used to prepare the front and back matter of your review manuscript. It is not necessary to follow the remaining structure. Structured reviews and meta-analyses should use the same structure as research articles and ensure they conform to the PRISMA guidelines.For this reason the manuscript would not be considered. Furthermore, it would be advisable to write the chemical formulas correctly with subscript, it would be the minimum. Furthermore, speculation about the future in terms of treatment and therapy seems excessive to me, especially if the revision rule has not been respected.
Author Response
Response to Reviewer 2 Comments
Title: Advances of Zinc Signaling Studies in Prostate Cancer
Manuscript ID: ijms-679303
Point 1: The authors declare that the document I have reviewed is a review. It has not been described in any way how the search for sources was made. In the "Instruction fro Authors" you can read:
Review manuscripts should comprise the front matter, literature review sections and the back matter. The template file can also be used to prepare the front and back matter of your review manuscript. It is not necessary to follow the remaining structure. Structured reviews and meta-analyses should use the same structure as research articles and ensure they conform to the PRISMA guidelines. 

Response 1: We thank the reviewer for the comments. We have carefully read the “Instruction for Authors” of the journal, and the article regarding the “PRISMA Statement” (PLoS Med, 6, e1000097, 2009) and the guidelines for “Systematic Reviews and Meta-Analyses”. Although we believe that our initially submission generally followed the guidelines, we added the “Contents” of the review in the revised manuscript. We also checked the “Checklist of items to include when reporting a systematic review or meta-analysis” described in the PLoS Med paper mentioned above and compared it point by point with our manuscript (see below for details).
Point 2: For this reason the manuscript would not be considered. Furthermore, it would be advisable to write the chemical formulas correctly with subscript, it would be the minimum. Furthermore, speculation about the future in terms of treatment and therapy seems excessive to me, especially if the revision rule has not been respected.
Response 2: We corrected the chemical formulas that we did not write the subscripts correctly. We checked our suggested predictions or expectations in the manuscript, and found most of them are the suggestions for future research directions. In addition, we indicated zinc reduction as a potential cancer biomarker and its potential application in cancer therapies.
We summarized our suggestions or perspectives of zinc/prostate cancer-related studies in future in the “Conclusion and Future Prospects” section. We feel that this should be an organized way for the readers to grasp the proposed future directions in this review. Most of these future directions were either directly or indirectly suggested in cited literature. Importantly, any suggestion in this manuscript is based on our review of the experimental data from cited papers.
“Point by point” comparison of the revised manuscript to conform the “Checklist of items to include when reporting a systematic review or meta-analysis” in the “PRISMA Statement” (PLoS Med, 6, e1000097, 2009).
Title: Identify the report as a systematic review, meta-analysis, or both.
The title of our manuscript is “Advances of Zinc Signaling Studies in Prostate Cancer”, which clearly indicates the article is a review for the “advances” of described study subject.
Provide a structured summary including, as applicable: background; objectives; data sources; study eligibility criteria, participants, and interventions; study appraisal and synthesis methods; results; limitations; conclusions and implications of key findings; systematic review registration number.
We have included or discussed all applicable requirements in the manuscript.
Describe the rationale for the review in the context of what is already known.
We have described the rationale of this review in the “Introduction” section. As we stated, “the aim of this review is to summarize the evidence in the most recent literature with a focus on the levels, biological activities, and relevant compounds of zinc in prostate cancer. Moreover, this review also deliberates about the potential of clinical applications to target zinc signaling in prostate cancer therapies.”
Provide an explicit statement of questions being addressed with reference to participants, interventions, comparisons, outcomes, and study design (PICOS).
In all sections, for particular questions and cited examples, we have provided necessary information regarding participants, interventions, comparisons, outcomes, and study design.
Indicate if a review protocol exists, if and where it can be accessed (e.g., Web address), and, if available, provide registration information including registration number.
This point is not applicable to our manuscript.
Specify study characteristics (e.g., PICOS, length of follow-up) and report characteristics (e.g., years considered, language, publication status) used as criteria for eligibility, giving rationale.
We provided necessary information as required for each cited case, study or report.
Describe all information sources (e.g., databases with dates of coverage, contact with study authors to identify additional studies) in the search and date last searched.
The reviewed information is based on published papers that are listed in the References.
Present full electronic search strategy for at least one database, including any limits used, such that it could be repeated.
The reviewed studies are all based on the searching results in the PubMed and corresponding original papers. The search strategy is very common to all researchers in the biomedical field and unnecessary to be described in the review.
State the process for selecting studies (i.e., screening, eligibility, included in systematic review, and, if applicable, included in the meta-analysis).
We selected published studies collected by the PubMed. Based on the topic of each section, the criteria of selecting papers are the relevance of the research contents of the published work.
Describe method of data extraction from reports (e.g., piloted forms, independently, in duplicate) and any processes for obtaining and confirming data from investigators.
We followed the general process of data extraction from cited reports. After finding the title of a paper is relevant to the topic of our review, we will carefully read the content of the paper, extract information from it, and also compare or integrate it with other related papers. Typically, if we do not find any critical problem or question, we will not contact the authors or investigators to confirm the data.
List and define all variables for which data were sought (e.g., PICOS, funding sources) and any assumptions and simplifications made.
When we extracted information related to our research topic, we considered and listed, if necessary, variables for which data were sought.
Describe methods used for assessing risk of bias of individual studies (including specification of whether this was done at the study or outcome level), and how this information is to be used in any data synthesis.
When reading through each cited paper, we generally evaluated whether the study was conducted in a stand and appropriate process. When comparing or integrating different studies, we also assessed whether any of the studies could have any bias in sample collection, data processing, etc.
State the principal summary measures (e.g., risk ratio, difference in means).
This criterium is generally not applicable to most of the sections in our manuscript. When reviewing the serum zinc levels of prostate cancer patients, we considered many factors to discuss the potential discrepancies among different studies.
Describe the methods of handling data and combining results of studies, if done, including measures of consistency (e.g., I2) for each meta-analysis.
This point is not applicable to our manuscript.
Specify any assessment of risk of bias that may affect the cumulative evidence (e.g., publication bias, selective reporting within studies).
Our focus in each cited paper is generally the trends of changes, such as high or low zinc levels, cell proliferation or death, cancer progression or regression, etc. We summarized the data based on the sound harmony among the research design, experimental procedure, data interpretation and overall conclusion in each cited paper.
Describe methods of additional analyses (e.g., sensitivity or subgroup analyses, meta-regression), if done, indicating which were pre-specified.
This is not applicable to our manuscript.
Give numbers of studies screened, assessed for eligibility, and included in the review, with reasons for exclusions at each stage, ideally with a flow diagram. For each study, present characteristics for which data were extracted (e.g., study size, PICOS, follow-up period) and provide the citations.
Most of cited reports are functional and experimental studies, describing promoted or inhibited cell proliferation, activated or repressed gene expression, etc. In these cases, these two criteria are not applicable. When comparing serum zinc levels among different reports, we provided a table to compare the data from these papers.
Present data on risk of bias of each study and, if available, any outcome-level assessment (see Item 12).
This point is not applicable to our manuscript.
For all outcomes considered (benefits or harms), present, for each study: (a) simple summary data for each intervention group and (b) effect estimates and confidence intervals, ideally with a forest plot.
This point is not applicable to our manuscript.
Present results of each meta-analysis done, including confidence intervals and measures of consistency.
This point is not applicable to our manuscript.
Present results of any assessment of risk of bias across studies (see Item 15).
This point is not applicable to our manuscript.
Give results of additional analyses, if done (e.g., sensitivity or subgroup analyses, meta-regression [see Item 16]).
This point is not applicable to our manuscript.
Summarize the main findings including the strength of evidence for each main outcome; consider their relevance to key groups (e.g., health care providers, users, and policy makers).
We provided a comprehensive summary of the review manuscript.
Discuss limitations at study and outcome level (e.g., risk of bias), and at review level (e.g., incomplete retrieval of identified research, reporting bias).
This point is not applicable to our manuscript.
Provide a general interpretation of the results in the context of other evidence, and implications for future research.
In the “Conclusion and Future Prospects”, we provided a general interpretation of zinc’s functional role in prostate cancer development, and also proposed future research directions.
Describe sources of funding for the systematic review and other support (e.g., supply of data); role of funders for the systematic review.
We provided the sources of funding supports.
Round 2
Reviewer 2 Report
I read again this systematic review. I still don't see the characteristics of a systematic review according to the PRISMA statement. Where are the keywords? The number of items found? How many have been excluded? Where's the diagram? I'm sorry but the review is interesting but not the methodology is not being transparent and isn't suitable to be published in a journal of this relief.
Round 3
Reviewer 2 Report
Overview is correct